# Conflagrations and the Wisdom of Aboriginal Sacred Knowledge

**David M. J. S. Bowman** 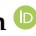

School of Biological Sciences, The University of Tasmania, Private Bag 55, Hobart, TAS 7001, Australia;
david.bowman@utas.edu.au; Tel.: +61-428-894-500

**Keywords:** Indigenous ecological knowledge; central Australia; fire disaster

Australian Aboriginal cultures are globally recognised for using patchy and low intensity fires to sustainably manage landscapes and promote biodiversity [1]. Following the disastrous 2019–2020 Australian fires, there has been increased discussion about whether such 'cultural' burning practices could have mitigated any subsequent fire disasters [2,3]. For example, an Indigenous colleague and I have advocated for the support of cultural burning programs [4]. My intention here is to draw attention to the striking similarities between an account of uncontrolled bushfire encoded in Aboriginal lore, and the bushfire 2019–20 crisis. This convergence of perspectives raises important questions, including how to appropriately combine Indigenous and scientific knowledge in the pursuit of sustainable fire management.

There are few recorded Aboriginal accounts of uncontrolled bushfires, possibly reflecting the extraordinary effectiveness of cultural burning practices in mitigating conflagrations on Aboriginal clan estates for over 50,000 years [5]. One exception is an evocative Aboriginal Dreaming narrative of a conflagration in the book *Journey to Horseshoe Bend* by THG Strehlow, first published in 1969 [6], recounting his boyhood experiences on the colonial frontier in 1922. Dreaming, or Dreamtime, is an English word to describe Aboriginal spiritual reality, and Dreaming narratives link the metaphysical realm to physical and social reality [7]. Strehlow grew up with the Arrernte people of central Australia in the early 20th century, and went on to become a renowned Australian anthropologist and Aboriginal linguist, with an exceptionally deep understanding of Arrernte culture.

In his book, Strehlow [6] recounts a sacred Arrernte story of firestorms lit by a 'malicious crow ancestor', who routinely set uncontrolled bushfire to terrify other Dreaming beings (pp. 193–196). These uncontrolled bushfires consumed the landscape with 'roaring tornadoes of leaping, billowing flames', and were so intense that trees were 'turned into swaying pillars of fire' and their roots reduced to ash. These Dreaming fires produced huge enveloping clouds of smoke, and the flames and heat had enormous impacts on ancestral wildlife where 'helpless marsupials, lizards and snakes died in their thousands'. While flocks of birds escaped, many were killed or injured. The red, black, and grey feathers possessed by modern-day birds are understood by the Arrernte people to symbolise red flames of the Dreaming conflagrations, black charred corpses, and grey ashes of the Dreaming birds incinerated by the inferno. Eventually, the rain ancestors subdued the firestorms, and the eternal heat and flames used by the destructive crow ancestor retreated underground to a sacred site. Without special and secret ceremonies by Arrernte people, the conflagrations could re-emerge, and this would 'magically induce the summer sun to increase its heat to such a dangerously high level that men and animals everywhere would be scorched to death'.

This Arrernte legend harmonises with other Aboriginal sacred knowledge that describes how birds can start landscape fires, no doubt encoding the behaviour of some bird species to transport burning sticks to intentionally spread fires [8]. The account has features that are strikingly similar to the 2019–20 Australian bushfire disaster that burned 7 million ha of *Eucalyptus* forest, much of it very severely [9], with catastrophic impacts on

wildlife and habitats [10,11]. These fires had extreme behaviour [12] with smoke pollution that released enormous quantities of greenhouse gases [13] and caused substantial human health impacts [14]. The Arrernte Dreaming story clearly stresses the need for humans to manage landscape fire to avoid catastrophe.

The incorporation of written and pictorial colonial accounts of landscapes and Aboriginal fire use has been transformative in understanding the importance of cultural burning in Australia [15]. A largely missing dimension in the historical reconstructions of landscapes and fire regimes is the foregrounding of Indigenous lore encoded in paintings, ceremonies, and oral traditions [16]. Nonetheless, there is increasing evidence of the close match between scientific and Indigenous knowledge of landscape evolution [17] biogeographic patterns [18], and wildlife biology [19]. A challenge ahead for fire scholars is finding effective and respectful collaborative means of incorporating the wisdom of Indigenous ecological knowledge into research and management programs [20–22]. This is an important step, because Indigenous knowledge can inform, reframe, challenge and substantiate scientific understanding of how fire shapes the world [23], leading to improved land management outcomes [3].

**Funding:** This research received no external funding.

**Conflicts of Interest:** The author declares no conflict of interest.

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
