# Peer review of "Conflagrations and the Wisdom of Aboriginal Sacred Knowledge"

_fire, doi:10.3390/fire4040088_

Round 1

Reviewer 1 Report

Thank you for the opportunity to review this Viewpoint. The topic is of great relevance to the current-day urgency of improving the way we manage and coexist with wildfires. While I am sympathetic to the argument made by the author, the actual argument is both half-baked and a missed opportunity.

1) Why base your argument on a 1969 white man's narrative about Aboriginal practices (a time when Aboriginal Australians still did not have all the same legal rights as white Australians), when a rich variety of Aboriginal voices as well as broader academic scholarship on indigenous cultural burning practices have emerged since?

2) While there may be few recorded Aboriginal accounts of uncontrolled bushfires (second paragraph), there is evidence of uncontrolled bushfire in many Aboriginal paintings (for example, among many, Clifford Possum Tjapaltjarri). This is an important point, as most Indigenous cultures (including Aboriginal Australian) historically have communicated orally and with art and not in written language.

2) It is a missed opportunity to reflect on significant related developments elsewhere. In particular, the recent passing of the prescribed fire bill by the California State Senate is a massive win for indigenous cultural fire stewards. Significant writing has also emerged from both the US and Canada, in some case with Australian comparative studies, about indigenous cultural burning practices in the past ten years, which are highly relevant to the argument made in this Viewpoint. I either recommend that the author incorporates these broader developments into a more substantive Perceptive piece, or that indigenous scholar(s) in Australia and North America are invited to write similar Viewpoint that are published alongside this piece (for example, Don Hankins, Amy Christiansen, Bill Tripp, Vanessa Cavanagh, Bhiamie Williamson). Otherwise, this Viewpoint feels like a continuation of sympathetic yet white people's views being expressed about indigenous cultural burning practices. The suggested alternatives feel both more inclusive and empowering of the people whose traditions and knowledge have sustained these cultural burning practices for millennia.

Author Response

Thank you for the opportunity to review this Viewpoint. The topic is of great relevance to the current-day urgency of improving the way we manage and coexist with wildfires. While I am sympathetic to the argument made by the author, the actual argument is both half-baked and a missed opportunity.

Thank you for the feedback.  I was not specifically mounting an argument, rather I was drawing attention to an extraordinarily interesting Aboriginal account of a mega fire and reflecting on this narrative.

  • Why base your argument on a 1969 white man's narrative about Aboriginal practices (a time when Aboriginal Australians still did not have all the same legal rights as white Australians), when a rich variety of Aboriginal voices as well as broader academic scholarship on indigenous cultural burning practices have emerged since?

I have clarified that Strehlow’s account relates to a journey he made in 1922, when the central Australian frontier was still very much in transition Strehlow is important (and controversial) figure in Australian Aboriginal studies, but there is no getting around he had an exceptional understanding of the Arrernte people (His father was a Lutheran missionary who went to central Australian in 1894, he was born there in 1908). The father and son have recorded an extraordinary amount of Aboriginal cultural knowledge.

To be clear, the point of this piece is to draw attention to this Arrernte narrative.  It is not a review of cultural burning, nonetheless in this piece I raise some of the complexities around the intersection of Indigenous knowledge and scientific perspectives.

  • While there may be few recorded Aboriginal accounts of uncontrolled bushfires (second paragraph), there is evidence of uncontrolled bushfire in many Aboriginal paintings (for example, among many,Clifford Possum Tjapaltjarri). This is an important point, as most Indigenous cultures (including Aboriginal Australian) historically have communicated orally and with art and not in written language.

Thank you, that is an important point and I have added this to my new final paragraph.

  • It is a missed opportunity to reflect on significant related developments elsewhere. In particular, the recent passing of the prescribed fire bill by the California State Senate is a massive win for indigenous cultural fire stewards. Significant writing has also emerged from both the US and Canada, in some case with Australian comparative studies, about indigenous cultural burning practices in the past ten years, which are highly relevant to the argument made in this Viewpoint. I either recommend that the author incorporates these broader developments into a more substantive Perceptive piece, or that indigenous scholar(s) in Australia and North America are invited to write similar Viewpoint that are published alongside this piece (for example, Don Hankins, Amy Christiansen, Bill Tripp, Vanessa Cavanagh, Bhiamie Williamson). Otherwise, this Viewpoint feels like a continuation of sympathetic yet white people's views being expressed about indigenous cultural burning practices. The suggested alternatives feel both more inclusive and empowering of the people whose traditions and knowledge have sustained these cultural burning practices for millennia.

The purpose of this piece was to introduce an astonishing coincidence between an Indigenous narrative encoded in a story and recent lived experience. It is beyond the scope of the piece to explore numerous matters arising, but hopefully this piece will stimulate further discussion.

To be clear my research is framed by who I am.  I am not ashamed of my commitment to science, yet I appreciate that other perspectives are essential to achieve sustainable and just stewardship of the Earth.

I think that is a marvellous idea to have a accompanying Indigenous perspective with the piece.  I think guidance about the incorporation of Indigenous knowledge into science is a much-needed step in the pathway to achieving sustainable land management. Bhiamie Williamson is chairing a working group I am on, and I think he would be an excellent first point of contact.

Reviewer 2 Report

In a world with increasing frequencies of megafires, which are burning with fire behaviors and effects modern fire agencies and researchers haven't seen before, it's refreshing to get a reminder, such as this viewpoint, that large fires aren't a new phenomena, and they can be elucidated through older traditional histories.  This provides a needed insight that, regardless of the frequency and application of cultural burning, that occasional large fires are still a noted possibility.  The large California wildfires of Summer 2021 are kind of a proxy for this: the size and magnitude of these fires far outstripped the fuels treatments and timber harvests emplaced over the last 25 years that were designed to mitigate wildfire damage.  Although effects were reduced, no-one was pondering the effects of a larger fire.

This viewpoint will provide a pretty strong argument for engaging with, and incorporating, indigenous histories regarding fire behavior over longer times than what our current records provide.

Author Response

Thank you for the strong endorsement of the piece.

Reviewer 3 Report

The paper needs some minor copyediting ("of" seem to be missing in the title and needs to be removed in the first sentence; one of the 'sacreds' in the first sentence of the 4th paragraph should be removed).

My main concern is with the ending.  I suggest that in the last paragraph "This Dreaming story..." be split off as a separate paragraph.  I think the author should be more explicit about his conclusion - a couple of sentences would be enough. Those of us who are not Australian are probably not reading the subtext that a native Australian would automatically insert.  Without a richer connection, a more explicit explication of the moral, the story stays in the Dreamtime.  In brief, the piece asks for a stronger punchline.

Author Response

Thanks you for this feedback.  I have added a final paragraph that attempts to explain the broader implications of this short piece.

Round 2

Reviewer 1 Report

Thank you for the revisions, which have gone some way to improve this Viewpoint. The piece still unnecessarily sidelines Indigenous voices and recent knowledge about Indigenous cultural burning. I think this can be addressed relatively easily via the following two steps:

1) Provide a clear context setter. At the end of the first paragraph, insert the following, which is paraphrased from the author’s response:

“This piece builds on a particular Arrernte fire narrative to draw attention to the similarities between Indigenous fire narratives encoded in lore and recent lived experience in Australia. It is not a review of cultural burning. Nonetheless, the piece highlights some of the complexities around the intersection of Indigenous knowledge and scientific perspectives, which I hope will stimulate further discussion.”

2) In the last paragraph where you now point to developments and knowledge gaps, insert the following references to work by indigenous and non-indigenous scholars who have examined these issues in-depth:

After: “A largely missing dimension in historical reconstructions of landscapes and fire regimes is the foregrounding of Indigenous lore encoded in paintings, song, dance and oral traditions” cite:

- Eriksen, C and Hankins, D.L. 2014. "The Retention, Revival and Subjugation of Indigenous Fire Knowledge through Agency Fire Fighting in Eastern Australia and California, USA." Society & Natural Resources 27 (12): 1288-1303. https://doi.org/10.1080/08941920.2014.918226

After: “A challenge ahead for fire scholars is finding effective and respectful collaborative means of incorporating the wisdom of Indigenous ecological knowledge into research and management programs” cite:

- Williamson, B and Weir, J.K. 2021. "Indigenous peoples and natural hazard research, policy and practice in southern temperate Australia: an agenda for change." Australian Journal of Emergency Management 36 (4): 62-67. https://doi.org/10.47389/36.4.62

- Smith, W, Neale, T, and Weir, J.K. 2021. "Persuasion without policies: The work of reviving Indigenous peoples’ fire management in southern Australia." Geoforum 120: 82-92. https://www.sciencedirect.com/science/article/pii/S0016718521000233

- Christianson, A. 2014. "Social science research on Indigenous wildfire management in the 21st century and future research needs." International Journal of Wildland Fire 24 (2): 190-200. http://dx.doi.org/10.1071/WF13048

Author Response

Thank you for the revisions, which have gone some way to improve this Viewpoint. The piece still unnecessarily sidelines Indigenous voices and recent knowledge about Indigenous cultural burning. I think this can be addressed relatively easily via the following two steps:

Thanks for this helpful feedback, which has improved this short piece.

1) Provide a clear context setter. At the end of the first paragraph, insert the following, which is paraphrased from the author’s response:

“This piece builds on a particular Arrernte fire narrative to draw attention to the similarities between Indigenous fire narratives encoded in lore and recent lived experience in Australia. It is not a review of cultural burning. Nonetheless, the piece highlights some of the complexities around the intersection of Indigenous knowledge and scientific perspectives, which I hope will stimulate further discussion.”

I have added some additional explanation for the intent of the piece, including a reference to a paper I wrote with an Indigenous colleague advocating for cultural burning.

2) In the last paragraph where you now point to developments and knowledge gaps, insert the following references to work by indigenous and non-indigenous scholars who have examined these issues in-depth:

After: “A largely missing dimension in historical reconstructions of landscapes and fire regimes is the foregrounding of Indigenous lore encoded in paintings, song, dance and oral traditions” cite:

- Eriksen, C and Hankins, D.L. 2014. "The Retention, Revival and Subjugation of Indigenous Fire Knowledge through Agency Fire Fighting in Eastern Australia and California, USA." Society & Natural Resources 27 (12): 1288-1303. https://doi.org/10.1080/08941920.2014.918226

After: “A challenge ahead for fire scholars is finding effective and respectful collaborative means of incorporating the wisdom of Indigenous ecological knowledge into research and management programs” cite:

- Williamson, B and Weir, J.K. 2021. "Indigenous peoples and natural hazard research, policy and practice in southern temperate Australia: an agenda for change." Australian Journal of Emergency Management 36 (4): 62-67. https://doi.org/10.47389/36.4.62

- Smith, W, Neale, T, and Weir, J.K. 2021. "Persuasion without policies: The work of reviving Indigenous peoples’ fire management in southern Australia." Geoforum 120: 82-92. https://www.sciencedirect.com/science/article/pii/S0016718521000233

- Christianson, A. 2014. "Social science research on Indigenous wildfire management in the 21st century and future research needs." International Journal of Wildland Fire 24 (2): 190-200. http://dx.doi.org/10.1071/WF13048

I have added these citations (and one other) and tidied up the text.